# Ceftazidime/Avibactam and Meropenem/Vaborbactam for the Management of Enterobacterales Infections: A Narrative Review, Clinical Considerations, and Expert Opinion

**DOI:** 10.3390/antibiotics12101521

**Published:** 2023-10-09

**Authors:** Andrea Marino, Edoardo Campanella, Stefano Stracquadanio, Maddalena Calvo, Giuseppe Migliorisi, Alice Nicolosi, Federica Cosentino, Stefano Marletta, Serena Spampinato, Pamela Prestifilippo, Stefania Stefani, Bruno Cacopardo, Giuseppe Nunnari

**Affiliations:** 1Department of Clinical and Experimental Medicine, University of Catania, 95123 Catania, Italy; cacopard@unict.it (B.C.); giuseppe.nunnari1@unict.it (G.N.); 2Department of Biomedical and Biotechnological Sciences, University of Catania, 95123 Catania, Italy; s.stracquadanio@unict.it (S.S.); alice-nicolosi93@hotmail.it (A.N.); stefania.stefani@unict.it (S.S.); 3Unit of Infectious Diseases, Department of Clinical and Experimental Medicine, University of Messina, 98124 Messina, Italy; edo.campanella93@gmail.com (E.C.); serenaspampinato93@gmail.com (S.S.); 4U.O.C. Laboratory Analysis Unit, A.O.U. “Policlinico-Vittorio Emanuele”, Via S. Sofia 78, 95123 Catania, Italy; maddalenacalvo@gmail.com (M.C.); gpp.miglio@gmail.com (G.M.); 5Unit of Infectious Diseases, ARNAS Garibaldi Hospital, University of Catania, 95122 Catania, Italy; federicacosentino91@gmail.com; 6Department of Diagnostic and Public Health, Section of Pathology, University of Verona, 37124 Verona, Italy; stefano.marletta92@gmail.com; 7Intensive Care Unit, ARNAS Garibaldi Hospital, 95122 Catania, Italy; pamelaprestifilippo@gmail.com

**Keywords:** ceftazidime–avibactam, meropenem–vaborbactam, carbapenem-resistant *Enterobacterales*, carbapenemases, Gram-negative infections, new BL/BLICs

## Abstract

This comprehensive review examines the unique attributes, distinctions, and clinical implications of ceftazidime–avibactam (CAZ-AVI) and meropenem–vaborbactam (MEM-VAB) against difficult-to-treat *Enterobacterales* infections. Our manuscript explores these antibiotics’ pharmacokinetic and pharmacodynamic properties, antimicrobial activities, in vitro susceptibility testing, and clinical data. Moreover, it includes a meticulous examination of comparative clinical and microbiological studies, assessed and presented to provide clarity in making informed treatment choices for clinicians. Finally, we propose an expert opinion from a microbiological and a clinical point of view about their use in appropriate clinical settings. This is the first review aiming to provide healthcare professionals with valuable insights for making informed treatment decisions when combating carbapenem-resistant pathogens.

## 1. Introduction

The latest European Centre for Disease Prevention and Control (ECDC) surveillance report highlights a 5.9% prevalence of patients in Europe with at least one healthcare-associated infection (HAI), ranging from 2.9% to 10% across countries [1,2]. These data show a warning infectious disease rate. Specifically, 98,166 patients suffer from one or more HAI per day, while 3.8 million are affected by at least one HAI every year. Globally, 4.5 million HAI episodes were spread among intensive care units across Europe between 2016 and 2017. According to the reports, pneumonia reached 21.4%, while upper respiratory tract infections stood at 4.3%. 

Urinary tract infections (18.9%), surgical site infections (18.4%), and bloodstream infections (10.8%) followed these high percentages. Finally, gastrointestinal infectious diseases revealed an 8.9% value. Among the main aetiological agents, *Escherichia coli* (16.1%), *Staphylococcus aureus* (11.6%), *Klebsiella* spp. (10.4%), *Enterococcus* spp. (9.8%), *Pseudomonas aeruginosa* (8.0%), *Clostridioides difficile* (7.4%), coagulase-negative staphylococci (7.1%), *Candida* spp. (5.2%), *Enterobacter* spp. (4.4%), *Proteus* spp. (3.6%), and *Acinetobacter* spp. (3.2%) have the highest rates [1,2]. These microorganisms are frequently related to high antimicrobial-resistance rates, which lead to challenging therapeutical strategies. The most common resistance episodes regard third-generation cephalosporin (33.3%) among *Enterobacterales* such as *Klebsiella pneumoniae* (60.3%). Similarly, *Enterobacterales* reach a 60.2% carbapenem-resistance percentage, with *K. pneumoniae* having the highest rate (20.4%). Carbapenem resistance is also common in *P. aeruginosa* isolates (30.2%) and *Acinetobacter baumannii* strains (77.0%) [2]. Carbapenem-resistant *Enterobacterales* (CRE) represent one of the most concerning multidrug resistant (MDR) pathogens, among which *E. coli* and *K. pneumoniae* emerge as common aetiological agents. These two species differ in susceptibility profiles, which show carbapenem resistance as a frequent occurrence only among *K. pneumoniae* strains (73.7%) [2,3]. Otherwise, *E. coli* isolates account for a 1.1% carbapenem-resistance rate.

The elevated carbapenem resistance prevalence complicates critically ill patients, whose therapeutical options often include carbapenems as last-line agents against extended-spectrum β-lactamases (ESBL) and AmpC-producing bacteria. Following that premise, CRE isolates reached the high-risk MDR category within the World Health Organization (WHO) alert report [1,4,5]. The recent approval of new β-lactam-β-lactamase inhibitor combinations (βL-βLICs) is the most promising strategy against MDR Gram-negative microorganisms causing severe infections. These next-generation combinations include cephalosporins or carbapenems together with β-lactamase inhibitors. The βL-βLICs combination allows β-lactam antibiotics to recuperate their antimicrobial effects thanks to the β-lactamase inhibitor’s hydrolytic activity against β-lactamases. Among the βL-βLICs, ceftazidime–avibactam (CAZ-AVI) and meropenem–vaborbactam (MEM-VAB) are strongly diffused within the clinical practice as “older” β-lactam antibiotics (ceftazidime and meropenem) and innovative β-lactamase inhibitors (avibactam and vaborbactam) combinations. Despite their similar mechanisms of action, different features of pharmacological properties, microbial targets, and resistance episodes emerged. Clinical trials support the fundamental impact of βL-βLICs in the case of systemic infections, requiring reliable minimum inhibitory concentration (MIC) values to guide antimicrobial treatment [6]. 

The present narrative review aims to describe CAZ-AVI and MEM-VAB features or differences in managing *Enterobacterales* infections. The following paragraphs analyse both combinations in terms of activity spectrum, pharmacological properties, and clinical impact on antimicrobial therapy. 

## 2. Chemical Structure, Pharmacological Properties, and Resistance Mechanisms

### 2.1. Ceftazidime–Avibactam

#### 2.1.1. Chemical Structures and Activity Spectrum

Ceftazidime (CAZ)—(6*R*,7*R*,*Z*)-7-(2-(2-aminothiazol-4-yl)-2-(2-carboxypropan-2-yloxyimino) acetamido)-8-oxo-3-(pyridinium-1-ylmethyl)-5-thia-1-aza-bicyclo[4.2.0] oct-2-ene-2-carboxylate (molecular formula: C_22_H_22_ N_6_O_7_S_2_; molecular mass: 546.58 g/mol)—is a third-generation cephalosporin. Like other cephalosporins, CAZ’s activity arises from side chains attached to the cephem nucleus at positions 3 and 7. The methylpyridinium group at position 3 and the carboxypropyl-oxyimino group at position 7 provide the antipseudomonal activity, whilst the aminothiadiazole ring at position 7 is responsible for the activity against Gram-negative bacilli. CAZ, such as all β-lactam antibiotics, binds to a variety of PBPs. However, due to its chemical structure, it binds primarily to PBP-3 of Gram-negative bacteria, despite inhibitor activity against other PBPs, such as PBP-1a or PBP-1b in E. coli, producing filamentation and a small release of endotoxin [7], with spheroplast formation followed by bacterial rapid lysis [8,9,10].

Avibactam (AVI)—trans-7-oxo-6-(sulfoxy)-1,6-diazabicyclo[3.2.1]octan-2-carboxamide (molecular formula: C_7_H_10_N_3_O_6_S; molecular mass: 265.25 g/mol)—is a non-β-lactam β-lactamase inhibitor belonging to diazabicyclooctanes (DBOs). AVI acts by inactivating susceptible β-lactamases through the formation of a carbamate bond between AVI’s position 7 carbonyl carbon and the same active-site serine that participates in acyl bonding with β-lactam substrates. Unlike “suicide” inactivators (clavulanic acid, sulbactam, and tazobactam), the resulting adduct does not undergo hydrolysis but rather a deacylation process to regenerate both the enzyme and the inhibitor. Although it is inactive against class B enzymes (metallo-β-lactamases), AVI has a broad spectrum of activity, efficiently inhibiting Ambler class A (TEM1, CTX-M-15, KPC-2, KPC-3), class C (AmpC), and certain class D β-lactamases (OXA-10, OXA-48). In vitro studies showed that only 1 to 5 molecules of avibactam are enough to inhibit 1 β-lactamase molecule, in comparison with 55 to 214 molecules of tazobactam and clavulanic acid. However, KPC represents an exception, as the AVI-enzyme adduct is slowly desulphated, generating inactive products, which means that a higher concentration of AVI is required to maintain effectiveness against KPC [11,12,13,14]. Figure 1 reports the CAZ-AVI chemical structure.

#### 2.1.2. Pharmacological Properties

AVI plasma protein binding is approximately 8%, which allows up to 90% of the drug to exert its activity. AVI presents a moderate tissue distribution, with a volume of distribution at a steady state (Vdss) ranging from 15 L to 25 L, and it is transported to tissues, such as the liver, through uptake transporters like OATP1B1 and OATP1B3. Of note, drugs that inhibit OAT1 and OAT3 should not be administered with AVI. The drug is predominantly eliminated via renal excretion (the biliary excretory pathway is not relevant), with an elimination half-life of less than 3 h, and active tubular secretion is also involved in the elimination process. Similarly, CAZ is exclusively renally cleared as well, with a renal excretory amount of around 90% and a half-life of approximately 2.5 h. Therefore, caution is needed in dose selection for both drugs in patients with renal impairment. Vdss of CAZ is approximately of 17 L, comparable to that of AVI. The pharmacokinetics of AVI were not affected by the coadministration of CAZ. Additionally, CAZ exhibits low plasma protein binding (approximately 21%), which makes it unlikely that any unforeseen interactions will occur due to displacement of protein-bound fractions. The most effective PK/PD parameter for measuring the efficacy of AVI in combination with CAZ is the percentage of the dosing interval during which the free AVI levels are above a threshold concentration of 1 mg/L (*fT* > C_T_). This parameter aligns with the principal PK/PD index of CAZ, which is the percentage of time that the free concentration of the drug remains above MIC (%*f*T > MIC) [10,15,16]. As regards epithelial lining fluid (ELF) penetration, which still represents a point of debate, Dimelow and colleagues reported that in healthy volunteers, the ELF concentrations achieved with the approved dosage regimens of CAZ-AVI were comparable to the plasma PK/PD targets, suggesting their efficacy in treating infections in the lung. In detail, the study found that CAZ penetration into ELF follows a saturable Michaelis–Menten model, meaning that penetration is higher at low plasma concentrations but saturates at high plasma concentrations, specifically above 250 mg/L. For CAZ, the ELF concentration reaches a maximum at 45.4 mg/L, with a half-maximal ELF concentration achieved at a plasma concentration of 71.7 mg/L. AVI penetration into ELF is also somewhat nonlinear, described by a power model. While noncompartmental methods suggested an average ELF penetration of around 35%, the analysis revealed that ELF penetration can be as high as 47% at relevant plasma concentrations [17,18].

#### 2.1.3. Resistance Mechanisms

CAZ-AVI resistance mechanisms are complex and may be simultaneously mediated by multiple pathways in a single cell. Most cases of CAZ–AVI resistance are caused by mutated bla_KPC_ genes, such as bla_KPC-2_ and bla_KPC-3_ in *K. pneumoniae*. Amino acid substitutions (D179Y, L169P), mutations (T243M), and insertions (P-N-K insertion between positions 269 and 270 in a KPC-41, which is a KPC-3 variant) frequently occur within the conserved motif region of class A β-lactamases named the omega (Ω)-loop. Of note, mutated bla_KPC_ genes conferring CAZ-AVI resistance can result in reduced or abolished carbapenemase activity. Mutations in bla_CTX-M_ genes represent other causes of CAZ-AVI resistance, with possible future epidemiological significance since CTX-M is one of the most prevalent types of ESBL. CAZ-AVI is approved against OXA-48-producing *Enterobacterales*. However, CAZ-AVI resistance caused by mutations in bla_OXA_ genes has been detected after exposure to CAZ-AVI in vitro, with amino acid substitution (P68A and Y211S) producing a fivefold reduction in AVI potency. Porin mutations and efflux pumps activity (especially with other mechanisms associated) play a role in CAZ-AVI resistance too. Although the entry of CAZ into the periplasmic space is thought to be less dependent on major porins (e.g., OmpK35 and OmpK36) than the entry of carbapenems, T333N substitution in OmpK36 was shown to reduce the susceptibility to CAZ-AVI. Likewise, the role of OmpK35 inactivation in CAZ-AVI resistance was identified [11]. Figure 2 summarizes CAZ-AVI mechanisms of action (Figure 2A) and possible resistance mechanisms (Figure 2B).

### 2.2. Meropenem–Vaborbactam

#### 2.2.1. Chemical Structures and Activity Spectrum

Meropenem (MEM)—(4*R*,5*S*,6*S*)-3-{[(3*S*,5*S*)-5-(dimethylcarbamoyl) pyrrolidin-3-yl]sulfanyl}-6-[(1R)-1-hydroxyethyl]-4-methyl-7-oxo1-azabicyclo[3.2.0]hept-2-ene-2-carboxylic acid (molecular formula: C_17_H_25_N_3_O_5_S; molecular mass: 383.46 g/mol)—is a broad-spectrum group 2 carbapenem antibiotic, with intrinsic stability to hydrolysis with most noncarbapenemase β-lactamases (including AmpC enzymes and ESBLs of the TEM, SHV, and CTX-M families) due to the trans configuration of C_5_-C_6_ and the C-_6_(R)-hydroxyethyl substituent. As with other carbapenems, this synthetic derivative of thienamycin displays activity against Gram-positive and Gram-negative bacteria. Moreover, the pyrrolidinyl substituent, along with the trans orientations of the 6-hydroxyethyl moiety, enhances MEM bactericidal activity against Gram-negative bacilli. Furthermore, the 1β-methyl, 2-thiopyrrolidinyl substituent at C_2_ provides stability to dehydropeptidase-1 inhibitor, differently from imipenem. MEM can bind to several PBPs (at least three), varying between bacteria. Against Enterobacterales and P. aeruginosa, meropenem binds primarily to PBP-2, although it displays affinity for PBP-1a, PBP-1b, PBP-3, and even PBP-4. However, the inhibition of PBP-2 and the low affinity to PBP-3 is responsible for the ability to cause rapid bactericidal action without filamentation [7,14,19,20,21].

Vaborbactam (VAB)—{(3R,6S)-2-Hydroxy3-[2-(thiophen-2-yl)acetamido]-1,2-oxaborinan-6-yl} acetic acid (molecular formula: C_12_H_16_BNO_5_S; molecular mass: 297.14 g/mol)—is the first non-β-lactam boronic acid β-lactamase inhibitor, with a broad spectrum of activity against various serine β-lactamases. Indeed, VAB was shown to inhibit various class A carbapenemases (KPC-2, KPC-3, KPC-4, BKC-1, FRI-1, and SME-2), class A ESBLs (CTX-M, SHV, and TEM), and class C cephalosporinases (CMY, P99), but it is practically inactive against metallo-β-lactamases and, unlike AVI, does not show appreciable inhibitory activity against class D carbapenemases (OXA-48-like). Due to the boron atom, VAB mimics the carbonyl carbon of the β-lactam ring. The covalent adduct deriving from the interaction between active-site serine of β-lactamases and the boronate moiety mimics the tetrahedral transition state on the acylation/deacylation pathway, resulting in rapid enzyme deactivation. The covalent bond is reversible, and VAB is not hydrolysed during the reaction. In contrast to AVI, against KPC, VAB is slowly desulphated, resulting in inactive products; the 2-thienyl acetyl group of VAB increases inhibitory potency against KPC-producing bacteria [14,22]. MEM and VAB display comparable pharmacokinetic profiles without drug–drug interactions between them. Figure 3 reports MEM-VAB chemical structure.

#### 2.2.2. Pharmacological Properties

The average plasma protein binding of MEM is very low, approximately 2%, while it is around 33% for VAB, a much higher percentage compared with other β-lactamase inhibitors. Both drugs are primarily renally excreted, exceeding the glomerular filtration rate due to active tubular secretion. However, while MEM presents some levels of nonrenal elimination (by dipeptidases, by nonspecific degradation, or by faecal elimination), VAB does not, showing less plasma clearance in patients with decreasing renal function. When administered as a 3 h intravenous infusion every 8 h, meropenem shows a half-life of 1.22 h, whereas the half-life of vaborbactam is 1.68 h. The Vdss of MEM and VAB in healthy adults is approximately 21 L for both drugs, expressing good penetration in tissues and body fluids. Furthermore, the peak plasma concentration (Cmax) and the area under the plasma concentration–time curve (AUC) values increase in a dose-related manner for both. The PK-PD parameters that were found to be the most effective in describing the antibacterial activity of MEM-VAB were percentage of the dosing interval during which free drug levels remain above the MIC (%*f*T > MIC) and the 24 h free VAB AUC/MEM-VAB MIC ratio, respectively. According to these data, the administration of 2 g MEM and 2 g VAB every 8 h by 3 h infusion is sufficient to kill bacteria and prevent the resistance of KPC-producing carbapenem-resistant strains of *Enterobacterales* with a MEM-VAB MIC of up to 8 mg/L [14,22,23]. As regards ELF penetration, following the administration of 2 g of MEM and 2 g of VAB as 3 h intravenous infusions, the average and median ratios of ELF concentrations to unbound plasma concentrations were 65/59% and 79/72%, respectively, following the third dose. It is worth noting that the ratio of the beta-lactamase inhibitor in alveolar macrophages (AM) ranged from 6 to 258% within the first 8 h after dosing. These findings suggest that when both MEM and VAB are administered at doses of 2 g every 8 h, they may attain effective concentrations in the ELF [24].

#### 2.2.3. Resistance Mechanisms

Emergence of MEM-VAB-resistant strains is reported worldwide. Studies on KPC-producing *K. pneumoniae* show that VAB, just like MEM, can cross the outer membrane through both OmpK35 and OmpK36 porins. The latter, however, appears to play a more important role for VAB, as the inactivation of OmpK35 was associated with a much smaller effect than the inactivation of OmpK36 or both. GD134-135 insertion is the most frequent mutation identified in the conserved L3 loop of ompK36. Among *Enterobacterales*, efflux pump systems are also implicated in reduced carbapenem susceptibility, in particular the AcrAB-TolC system. In a study conducted by Lomovskaya et al. [25], a bigger reduction in MEM-VAB potency was observed in KPC-producing strains lacking both porins and overexpressing AcrAB. Nevertheless, upregulation of this pump only exhibited a minimal effect on MEM-VAB potency because it does not produce a measurable effect on the MIC of MEM and because VAB is a poor substrate for AcrAB-TolC. Increased MIC to MEM-VAB was reported by Sun et al. [26] in KPC-Kp strains with inactivation or diminished function of OmpK36 along with overexpression of KPC. Intracellular transposition of Tn4401, an increase in the number of copies of bla_KPC_ per plasmid, or an increase in the number of KPC-carrying plasmids per cell and insertional inactivation of the *rep*A2 gene represent the more important mechanisms behind the increased Bla_KPC_ copy number. In conclusion, as shown by Zhou et al. [27], strains showing a complete inactivation of porins in combination with increased expression of bla_KPC_ and *acr*AB genes were associated with the highest MIC for MEM-VAB [5,14,25]. Figure 4 summarizes MEM-VAB mechanisms of action (Figure 4A) and possible resistance mechanisms (Figure 4B).

## 3. Susceptibility Testing

According to EUCAST [28], broth microdilution is the gold-standard method among susceptibility testing procedures for CAZ-AVI and MEM-VAB due to its high sensitivity and reliability. Unfortunately, this technique is not easy to manage in a diagnostic laboratory routine. For this reason, it was necessary to expand susceptibility tests through alternative methods. Particularly, gradient test strips are a valuable alternative technique for gathering reliable MIC data. Furthermore, some automated systems currently provide CAZ-AVI and MEM-VAB susceptibility profiles.

Several studies were performed to evaluate gradient test reliability in testing *Enterobacterales* CAZ-AVI susceptibility. A multicentre analysis involved 83 *Enterobacterales* strains comparing Vitek 2 and EUCAST broth microdilution to test CAZ-AVI susceptibility. The investigation reported an essential agreement of more than 97% and a categorical agreement of 100% between the two techniques. No very major errors or major errors were recorded. As a result, Vitek 2 represents a considerable susceptibility testing method for CAZ-AVI [29].

Jean et al. [30] produced a multicentric clinical evaluation of the gradient test versus EUCAST broth microdilution in testing *Enterobacterales* MEM-VAB susceptibility. More than 600 strains were processed through the two methods, which showed an essential agreement of 92.4% and a categorical agreement of 99.2%. No major errors or very major errors were detected. Consequently, the gradient test could be used as a reliable method to test *Enterobacterales* susceptibility to MEM-VAB. However, species-specific agreement rates were also calculated. Particularly, a very low percentage (34.3%) was reported for *Proteus mirabilis*, suggesting that the gradient test method should be avoided to test its susceptibility to MEM-VAB.

As regards automated systems, multicentre evaluations were performed to test *Enterobacterales* susceptibility by comparing broth microdilution to Vitek 2. For instance, more than 1000 Gram-negative isolates were included in a multicentre study by Humphries et al. [31], who demonstrated few major errors or very major errors between the two methods. According to this investigation, Vitek 2 results closely correlated with broth microdilution MIC data with an essential agreement and a categorical agreement >90%. Therefore, the Vitek 2 system can be validated as a valuable and automated method to test *Enterobacterales* CAZ-AVI susceptibility [31]. Dwivedi et al. [32] provided a multicentre evaluation of MEM-VAB susceptibility testing. This investigation involved large groups of *Enterobacterales* strains, which were tested through both the Vitek 2 automated system and EUCAST broth microdilution. MEM-VAB performance revealed an essential agreement of 97.3% and a categorical agreement >90% to the broth microdilution. In addition, no major errors or very major errors were recorded, confirming the absence of false resistance or false susceptibility trends. In conclusion, literary data support gradient tests and automated systems to test CAZ-AVI and MEM-VAB in laboratory practice. According to the above studies, both these techniques appear to be valid methods to gather reliable MIC results [29].

## 4. Clinical Data and Considerations

The optimal treatment of KPC-producing *Enterobacterales* infections is currently not well defined, nor is the question of whether the combination therapy would be superior to monotherapy, as there are a small number of well-conducted randomized clinical trials (RCTs). Hence, the therapeutic choice is mainly based on clinical experience.

In the context of carbapenemase-producing *Enterobacterales* (CPE), numerous cohort studies reported clinical success rates exceeding 65–70% when using CAZ-AVI for severe infections caused by KPC- or OXA-48-like-producing *Enterobacterales* [33,34,35,36,37] and with MEM-VAB for severe infections caused by KPC-producing *Enterobacterales* [38].

### 4.1. CAZ-AVI

CAZ-AVI received approval in Europe in 2016 [39]. It is approved for the treatment of adults with various conditions, including complicated urinary tract infection (cUTI), complicated intra-abdominal infection (cIAI), and hospital-acquired pneumonia/ventilator-associated pneumonia (HAP/VAP), including cases where bacteraemia is associated with these infections (bacteraemia can occur as a primary bloodstream infection (BSI) or as secondary to acute systemic infections). Additionally, it is indicated for treating infections caused by aerobic Gram-negative bacteria when treatment options are limited [39].

In recent years, real-world studies demonstrated the effectiveness of CAZ-AVI against MDR Gram-negative bacterial infections, particularly CRE, in both the United States (USA) and Europe. However, these studies often had limitations, such as focusing on specific infection types or particular countries [39,40].

CAZ-AVI treatment outcomes in CRE infections vary by infection type, with pneumonia and mechanical ventilation increasing the risk of treatment failure. While CAZ-AVI effectively penetrates lung tissue, the optimal dosage for pneumonia patients, especially those on ventilation, requires further study.

The findings from multiple RCTs [40,41,42] and a meta-analysis [43] provide evidence supporting the effectiveness of CAZ-AVI compared to carbapenems in terms of reducing mortality and achieving clinical improvement endpoints [2]. This holds true for cUTIs, complicated intra-abdominal infections (cIAIs) (when used with metronidazole), and nosocomial pneumonia, even when focusing on *Enterobacterales* that produce ESBL or AmpC enzymes [44]. However, concerns exist regarding the widespread use of CAZ-AVI for these common conditions. Notably, there is a risk of reduced effectiveness against *Enterobacterales* that produce KPC enzymes due to mutant selection [45,46], and there is a lack of real-world data demonstrating a lesser impact on commensal microbiotas compared to carbapenems.

One of the rising problems in the treatment of KPC-producing strains is resistance to CAZ-AVI. Resistance typically occurs after 10 to 19 days of drug exposure and often manifests during treatment of recurrent infection [46,47]. Ackley et al. [48] reported that 3 of 15 (20%) patients developed resistance within 90 days, with 2 patients experiencing treatment failure after approximately 2 weeks of CAZ-AVI monotherapy. Despite this, resistance appears to be able to develop even without previous treatment with CAZ-AVI. Mutations in the active site of the enzyme are mainly responsible for this phenomenon, with D179Y mutation being the most frequent. This mutation was ascribed to a 16-fold CAZ-AVI MIC increase [49,50,51]. The assumption is that KPC Ω-loop substitution results in stabilizing interactions (e.g., hydrogen bonds), prolonging CAZ binding at the active site, and thus preventing the inhibitory activity of AVI.

Real-world studies further support the efficacy of CAZ-AVI. For instance, Soriano et al. [52] assessed the real-world clinical outcomes of hospitalized adult patients who received at least one dose of CAZ-AVI for approved indications in routine clinical practice. The study enrolled 569 hospitalized patients; the main indications for treatment were HAP/VAP, cUTI, BSI, and cIAI. The median duration of CAZ-AVI administration was 9 days, with the majority receiving 2g/0.5g doses three times daily. *Klebsiella pneumoniae* was the most common pathogen, followed by *Pseudomonas aeruginosa*, *Escherichia coli*, and *Enterobacter cloacae*. The most common mechanisms of resistance were KPC, OXA-48, ESBL, and metallo-beta-lactamases (MBL). Notably, 7.8% of isolates tested were found resistant to that antibiotic before its initiation (mostly *K. pneumoniae*). Treatment success was achieved in 77.3% of patients, with the highest success rate observed in cUTI and BSI. The 60-day mortality rate was highest in HAP/VAP patients and the lowest in cUTI patients. Factors associated with decreased clinical success included older age (>80 years), infections caused by “other” Gram-negative bacteria, cIAI and HAP/VAP indications, and concomitant colistin use.

Favourable results were also shown by Jorgensen et al. [53], who conducted a retrospective cohort study involving 203 patients receiving CAZ-AVI treatment for at least 72 h at six medical centres in the USA. In their study, the most common indications for treatment were respiratory infections, UTIs, and IAIs. CRE were the most identified Gram-negative bacteria, followed by *Pseudomonas* species. Clinical failure was 29.1% and the 30-day mortality rate was 17.2%; no CAZ-AVI resistance was detected on retest after treatment. The patients with primary bacteraemia or a respiratory tract infection experienced the highest rates of clinical failure. Additionally, when it came to 30-day mortality, these same groups of patients saw higher rates with primary bacteraemia and respiratory tract infections.

In addition, Calvo-Garcia et al. [54] conducted a retrospective observational study between January 2016 and October 2018, in which CAZ-AVI was evaluated as a treatment option for patients with CRE infections. The study included 63 in-patients treated with CAZ-AVI. The most frequent source of infection was intra-abdominal, and *K. pneumoniae* was the most common CRE isolated. A high clinical cure rate (74.6%) was achieved. However, microbiological cure rates were slightly lower (55.6%). Factors such as ICU admission and the presence of bacteraemia were associated with reduced treatment effectiveness and increased infection recurrence at 90 days.

Lastly, Yang et al. [55] performed a meta-analysis that aimed to compare the effectiveness and safety of CAZ-AVI versus polymyxins in the treatment of infections caused by CRE. The authors conducted a comprehensive literature search, and the studies they found involved a total of 1111 patients. The researchers assessed various outcomes, including 30-day mortality, clinical success, bacterial eradication, and nephrotoxicity. Their results indicated that CAZ-AVI was associated with a lower 30-day mortality rate compared to polymyxins. Additionally, patients treated with CAZ-AVI had a higher clinical cure rate and experienced lower nephrotoxicity. However, there was no significant difference in bacterial eradication between the two treatment groups, which may be due to small sample size of the study.

As regards paediatric populations, CAZ-AVI was approved for paediatric patients ≥3 months with the same adult indications. Furthermore, there are several case reports and one case series describing its safety and efficacy in neonates [36].

### 4.2. MEM-VAB

RCTs investigating the use of MEM-VAB against CRE are limited.

The TANGO I study, a phase III multicentre double-blind randomized clinical trial, led to the approval of MEM-VAB for cUTIs. Nevertheless, the limited occurrence of CRE and the specific population with cUTI/AP restrict the study’s suitability for effectively assessing the role of MEM-VAB in different indications [56].

These issues were explored in the TANGO II, a randomized multicentre open-label phase 3 study. Comparing MEM-VAB monotherapy with best available therapy (BAT) in patients with confirmed or suspected CRE infections (cUTI/AP, HABP/VABP, bacteraemia, or cIAI), the study demonstrated more favourable clinical and microbiological responses in the MEM-VAB-treated group when compared to patients treated with BAT, which may include polymyxins, carbapenems, aminoglycosides, or tigecycline, either alone or in combination. The test-of-cure follow-up revealed that patients treated with MEM-VAB had a higher rate of clinical cure than those treated with BAT (57.1% vs. 26.7%), even in immunocompromised patients (70% vs. 0%). Moreover, although not statistically significant, survival was also superior in the MEM-VAB group, with a 28-day mortality rate of 17.9% compared to 33.3% in the BAT group. Since the control group included the use of amino-glycosides and/or colistin, renal adverse effects were more frequent in this group. It is worth noting that no evidence of resistance was found among patients receiving MEM-VAB, although one isolate (3.1%) had a fourfold increase in MIC from 0.25 to 1 µg/mL, remaining within the susceptibility range. By contrast, in the BAT group, 6.7% of isolates developed a >fourfold increase in MIC [57].

A post hoc analysis of the TANGO II trial conducted by Bassetti et al. [58], despite some limitations, showed that in patients with serious CRE infections without prior antimicrobial failure, MEM-VAB was superior to BAT when both strategies were employed as the first line of treatment.

Published real-world experiences showed good results and supported the findings from RCTs.

Shields et al. [38] presented a single-centre, real-world, prospective, observational study of 20 patients (70% of them in ICU) with CRE infections who were treated with MEM-VAB, finding thirty-day clinical success and survival rates of 65% and 90%, respectively. Moreover, they also provided meaningful data regarding the efficacy of MEM-VAB against CRE pneumonia, an underrepresented infection type in TANGO II.

Alosaimy and colleagues [59] conducted a multicentre retrospective cohort study which encompassed 126 patients with suspected or confirmed infections caused by MDR Gram-negative bacteria. CRE accounted for 78.6% of infections, with a smaller number attributed to other pathogens, including two cases of *A. baumannii* and eight cases of *P. aeruginosa*. The primary sources of infection varied, with a notable proportion originating from the respiratory tract (38.1%) and the intra-abdominal region (19%). Thirty-day mortality and recurrence occurred in 18.3% and 11.9%, respectively, comparable to results from the TANGO II trial. This is particularly noteworthy due to the high-risk profile of the patients in the study, who exhibited common risk factors for MDR infections, underscoring the significance of these data, especially given that patients with these risk factors are typically excluded from RCTs. Notably, receiving early treatment with MEM-VAB (within 48 h of symptom onset) was independently associated with a more favourable clinical outcome, as indicated by multivariable analysis (adjusted odds ratio, 0.277; 95% confidence interval, 0.081–0.941).

In a retrospective observational cohort study conducted across 12 Italian hospitals [60], 37 patients with infections caused by KPC-producing *Klebsiella pneumoniae* (KPC-Kp) were enrolled, including 23 BSIs, 10 low respiratory tract infection (LRTIs), 2 cUTIs, 1 ABSSSI, and 1 IAI. Notably, 70% of these cases were diagnosed in the intensive care unit (ICU). Clinical cure was achieved in 28 of the 37 cases, but 3 of these patients (all BSIs) experienced a recurrence after discontinuing MEM-VAB. However, isolates remained susceptible to MEM-VAB, and microbiological and/or clinical cures were eventually achieved through retreatment with MEM-VAB in combination with colistin (in two cases) or fosfomycin (in one case). Nine patients died (24.3%), and all nine had either BSIs or LRTIs. It is worth noting that because patients received MEM-VAB through a compassionate-use program, there was a significant time gap between infection onset and the initiation of MEM-VAB, with a median delay of 5 days. Although limitations are given by the sample size, it is important to note that six out of the nine patients who passed away in the hospital had initiated MEM-VAB treatment ≥48 h after the initial culture, aligning with data produced by Alosaimy et al. [59].

There are some reports in the medical literature regarding the utilization of MEM-VAB in conjunction with aztreonam for the treatment of MBL-producing *Enterobacterales*. Tiseo et al. documented two cases of infections caused by New Delhi metallo (NDM)-β-lactamase-producing Klebsiella pneumoniae (Kp) that were successfully treated with this particular combination [61]. In a separate case series, Belati et al. [62] provided insights into the use of MEM-VAB plus aztreonam to treat infections caused by CAZ-AVI-resistant *Klebsiella pneumoniae* (Kp), including two cases sustained by NDM-Kp. However, it is crucial to note that the author emphasized the significance of knowing the local epidemiology of MDR Gram-negative bacteria, as the use of the MEM-VAB plus aztreonam combination is discouraged in settings where OXA-like carbapenemases are frequently reported.

### 4.3. CAZ-AVI vs. MEM-VAB

Some studies assessed the in vitro activity of MEM-VAB and CAZ-AVI against isolates of Gram-negative bacilli. Rogers et al. [63] analysed genomes of 104 nonconsecutive KPC-Kp isolates and compared the in vitro antibiotic activity of CAZ-AVI, MEM-VAB, and imipenem/relebactam against genetically diverse KPC-Kp clinical isolates. The authors found that MICs for each agent were elevated against isolates harbouring IS5 mutations in ompK36 (by 2-, 4-, and 16-fold, respectively). However, median MICs for MEM-VAB were well below the current susceptibility breakpoint and, based on PK/PD modelling, the agent would be predicted to be effective even at the highest MICs identified in the study.

Hackel et al. [64] compared the in vitro activity of MEM-VAB against a collection of 991 isolates of KPC-positive *Enterobacterales*. In their analysis, MEM-VAB showed more potent in vitro activity compared to MEM alone, CAZ-AVI, tigecycline, ceftazidime alone, minocycline, polymyxin B, and gentamycin.

Clinical efficacy of novel βL-βLIC combinations in CPE infections was directly compared in only one retrospective study, by Ackley and colleagues [48]. The study primarily focused on infections resulting from KPC-producing *Enterobacterales* (comprising 72% of cases) and included critically ill patients comprising approximately half the study population. In the study, both CAZ-AVI (n = 105) and MEM-VAB (n = 26) demonstrated similar outcomes in terms of clinical and microbiological successes, duration of hospitalization, incidence of adverse events, and mortality. Notably, a difference emerged in terms of resistance development: CAZ-AVI led to three patients developing resistant strains, whereas no such cases occurred with MEM-VAB; however, due to the retrospective nature of the study and the sample size, it is not possible to draw clinical conclusions.

As much as can be gleaned from our clinical experience and the examined scientific literature, we can state that both combinations (MEM-VAB and CAZ-AVI) play a key role in the therapy of KPC-producing *Enterobacterales*. To date, there are no studies demonstrating the therapeutic superiority of one over the other in treating *Enterobacterales* infections.

Clinically significant differences concern MEM-VAB’s ability to also cover KPC subtypes in contrast to its lack of coverage against OXA-producing bacteria; a more favourable βL-βLICs ratio (1:1) could translate into a lower likelihood of developing resistance even in monotherapy and greater antibiotic potency due to the presence of the carbapenem (saturating a higher number of PBPs). In addition, MEM-VAB shows efficacy against anaerobic bacteria. CAZ-AVI can cover OXA-producing *Enterobacterales* while showing limited activity against KPC subtypes, and it has no activity against anaerobic bacteria. Additionally, it allows for a carbapenem-sparing therapy, which is considered fundamental in antibiotic stewardship programs [65,66]. Neither of the two combinations, on their own, exhibit activity against MBL-producing bacterial strains or MDR *A. baumannii* [67]. Figure 5 emphasizes the CAZ-AVI and MEM-VAB inhibitions targets, highlighting a comparison between the two combinations.

## 5. Expert Opinions

### 5.1. The Microbiological Point of View

CAZ-AVI generally shows a good susceptibility profile for KPC- and OXA-producing strains. However, some KPC variants could compromise its antimicrobial activity. On the other hand, MEM-VAB efficiently acts against KPC, including possible variants, exhibiting a resistance rate in the case of OXA-producing isolates.

These assumptions justify the possibility of testing MEM-VAB in the case of a CAZ-AVI resistance, matching the carbapenem-sparing strategy and preserving the carbapenem molecule. Therefore, a CRE susceptibility profile should include the possibility of selective reporting, enriching the MIC panel data after a consistent collaboration between infectious disease clinicians and laboratory personnel. Unfortunately, the literature data document that the contemporary presence of KPC-3 variants and OXA-48 genes among *K. pneumoniae* strains leads to CAZ-AVI and MEM-VAB simultaneous resistance [68]. These considerations highlight the importance to always search both resistance markers after a complete susceptibility profile analysis. In our opinion, the epidemiological context could also support information on all phenotypic susceptibility testing. Specifically, we currently operate among a high-resistance prevalence area that has a prevalence of National Health Institute-analysed carbapenem-resistance markers. The gathered reports showed a value of 2.2% of OXA-48-producing *K. pneumoniae* among Italian regions during 2022. Otherwise, the presence of KPC-producing *K. pneumoniae* strains reached an 80.6% value during the same period, with small KPC variants included [69]. The reports explain how CAZ-AVI was a significant option against all KPC-producing strains, while MEM-VAB provided coverage in cases where CAZ-AVI might fail, particularly in the presence of KPC variants, but not against OXA-producing strains.

### 5.2. The Clinical Point of View

The treatment landscape for KPC-producing *Enterobacterales* infections remains challenging due to the limited availability of well-conducted randomized clinical trials. The therapeutic approach is primarily guided by clinical experience, real-world evidence, and epidemiological/microbiological data. CAZ-AVI has demonstrated efficacy in various clinical settings, including cUTIs, cIAIs, and HAP/VAP. Real-world studies consistently show positive outcomes, with clinical success rates exceeding 65–70% for severe infections caused by KPC- or OXA-48-like-producing *Enterobacterales*. However, concerns exist regarding the development of resistance, which can occur after prolonged exposure to the drug. Further research is needed to optimize dosing, especially in pneumonia patients on mechanical ventilation. MEM-VAB, on the other hand, shows promise in the treatment of CRE infections. The TANGO II study and subsequent real-world experiences highlighted its clinical and microbiological efficacy, even in immunocompromised and severely ill patients.

Notably, resistance development appears to be less common with MEM-VAB compared to CAZ-AVI, even though these data could be ascribed to the fact that CAZ-AVI has been in clinical use for a longer time than MEM-VAB [68].

Comparing the two treatments, there is no clear evidence of one being superior to the other in treating *Enterobacterales* infections. Each has its strengths and limitations. CAZ-AVI covers OXA-producing bacteria and offers carbapenem-sparing therapy, while MEM-VAB provides broader coverage against KPC subtypes and anaerobic bacteria and may have a lower risk of resistance development.

The choice between the two drugs should be tailored to the specific clinical scenario and local epidemiology. It is also important to consider that the options are not mutually exclusive, as CAZ-AVI could be an excellent choice for de-escalation therapy from MEM-VAB. In fact, in critically ill patients and in an MDR epidemiological setting, the consideration of MEM-VAB as the initial empirical treatment, if the microbiological isolate is found to be susceptible, could be a viable option to save the carbapenem switching to CAZ-AVI.

The microbiology laboratory should support the clinician’s choice and therapy adjustment. As recently published [70], some *K. pneumoniae* strains may be resistant to both CAZ-AVI and MEM-VAB due to the contemporary presence of KPC-3 and OXA-48. Although the automatized systems do not discriminate among the KPC variants, in this scenario, it is of great importance at least to search for both the resistance genes and always demand in vitro susceptibility testing.

In summary, both CAZ-AVI and MEM-VAB play pivotal roles in the management of KPC-producing *Enterobacterales* infections. Further research and real-world data are essential to refine treatment strategies and optimize patient outcomes in this challenging clinical context.

## 6. Key Messages

I.The optimal treatment for KPC-producing *Enterobacterales* infections is not well defined due to a lack of well-conducted RCTs, making treatment decisions primarily reliant on clinical experience.II.CAZ-AVI has demonstrated effectiveness in treating severe infections caused by KPC- or OXA-48-like-producing *Enterobacterales*, with high clinical success rates.III.CAZ-AVI’s efficacy varies by infection type: pneumonia and mechanical ventilation may increase the risk of treatment failure due to penetration issues. Optimal dosing for pneumonia patients, especially those on ventilation, requires further studies.IV.Concerns exist regarding CAZ-AVI’s effectiveness against KPC enzyme subtypes and KPC overexpression. Resistance to CAZ-AVI may develop after prolonged drug exposure, posing a challenge in the treatment of recurrent infections.V.MEM-VAB shows promise in treating CRE infections thanks to its microbiological potency and PK characteristics.VI.MEM-VAB is highly effective against KPC-producing strains, being active even against KPC subtypes. It has no effect against OXA-48 producing isolates.VII.Resistance development appears less common with MEM-VAB compared to CAZ-AVI, but further research is needed to understand long-term resistance patterns.VIII.There is no clear evidence of one drug being superior to the other in treating Enterobacterales infections. The choice should be tailored to the specific clinical scenario and local epidemiology.IX.Microbiology laboratories play a crucial role in supporting treatment decisions by providing susceptibility profiles, and clinicians should consider local resistance patterns when choosing between CAZ-AVI and MEM-VAB.X.Both CAZ-AVI and MEM-VAB are essential in managing KPC-producing Enterobacterales infections, and further research is needed to optimize treatment strategies in this challenging context.

## 7. Materials and Methods

A comprehensive literature search was conducted to identify relevant studies comparing the efficacy, safety, and clinical outcomes of ceftazidime–avibactam and meropenem–vaborbactam. The search strategy was implemented using online databases, including PubMed/MEDLINE, Scopus, Web of Science, and relevant clinical trial registries. The search was not restricted by language or publication date and covered articles up to the cutoff date of September 2023. The following keywords and MeSH terms were used: “Ceftazidime-Avibactam”, “Meropenem-Vaborbactam”, “Combination therapy”, “Antibacterial agents”, “Infection control”, “Clinical outcomes”, “Randomized controlled trials”, “Systematic review”. Studies were included in this narrative review if they met the following criteria: comparative studies (including randomized controlled trials, observational studies, and systematic reviews/meta-analyses) that directly compared ceftazidime–avibactam and meropenem–vaborbactam in the treatment of various bacterial infections; studies reporting clinical outcomes, including but not limited to microbiological eradication rates, clinical cure rates, safety profiles, and adverse events associated with the use of either drug combination; and studies involving human subjects of all age groups.

The search strategy had no time limits or language restrictions. We screened the articles by title and abstract in full text if relevant. To complement the evidence from the peer-reviewed literature, we searched for papers, abstracts, research reports, and case studies on the web. Conference abstracts were checked to avoid duplication of the peer-reviewed literature. In the case of duplication, the full text article was preferred. Three reviewers independently searched and reviewed the studies. Any discrepancies were resolved by the other two reviewers. After an initial screening of titles and abstracts of published articles, the reviewers evaluated full articles to assess eligibility for each study’s inclusion in this narrative review. A study was included if it was likely to provide valid and valuable information according to the review’s objective.

Studies with insufficient data or incomplete reporting of relevant outcomes, animal studies, in vitro experiments, and nonoriginal research articles such as editorials, commentaries, and case reports were excluded.

## Figures and Tables

**Figure 1 antibiotics-12-01521-f001:**
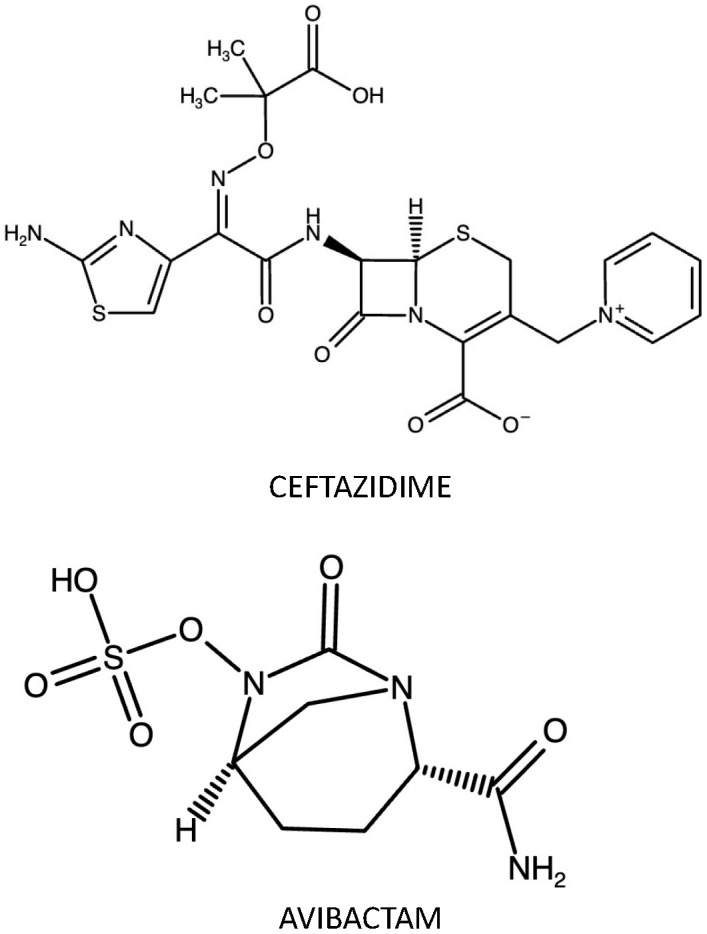
CAZ-AVI chemical structure. Made with https://www.ebi.ac.uk/chembl/ (accessed on 20 Semptember 2023).

**Figure 2 antibiotics-12-01521-f002:**
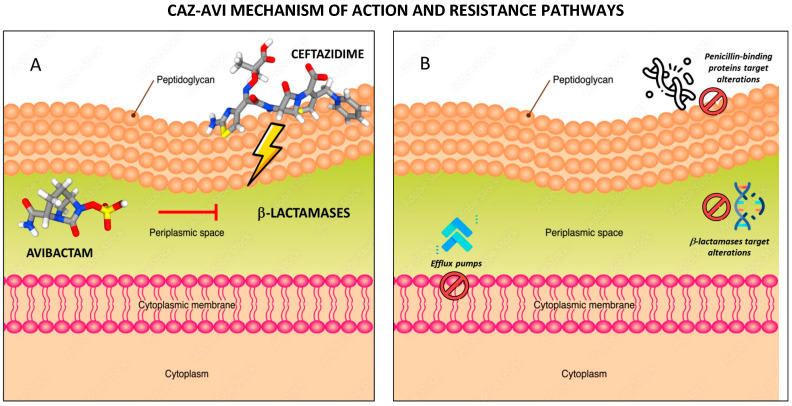
CAZ-AVI mechanism of action (**A**) and resistance mechanisms (**B**). Created using image databases https://stock.adobe.com/it/ (accessed on 15 September 2023) and https://pubchem.ncbi.nlm.nih.gov (accessed on 15 September 2023).

**Figure 3 antibiotics-12-01521-f003:**
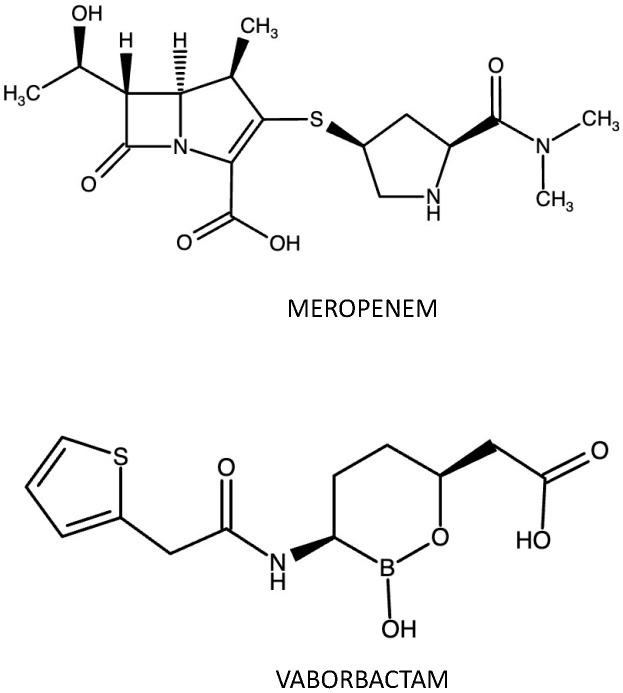
MEM-VAB chemical structure. Made with https://www.ebi.ac.uk/chembl/ (accessed on 25 September 2023).

**Figure 4 antibiotics-12-01521-f004:**
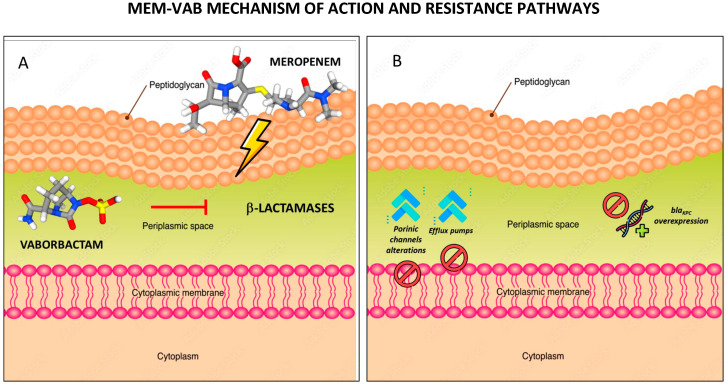
MEM-VAB mechanism of action (**A**) and resistance mechanisms (**B**). Created using image databases https://stock.adobe.com/it/ (accessed on 15 September 2023) and https://pubchem.ncbi.nlm.nih.gov (accessed on 15 September 2023).

**Figure 5 antibiotics-12-01521-f005:**
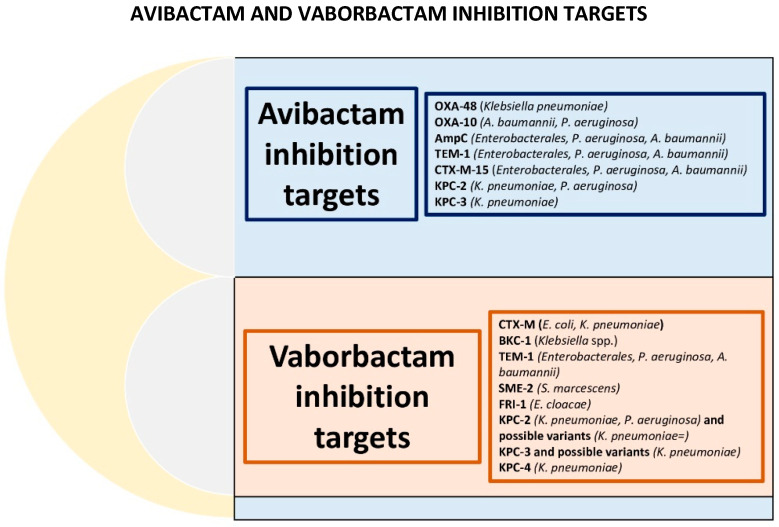
Schematization of avibactam and vaborbactam inhibition gene targets.

## Data Availability

Not applicable.

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
