# Peer review of "Ceftazidime/Avibactam and Meropenem/Vaborbactam for the Management of Enterobacterales Infections: A Narrative Review, Clinical Considerations, and Expert Opinion"

_antibiotics, 2023, doi:10.3390/antibiotics12101521_

Round 1

Reviewer 1 Report

Line 22: Please restructure the sentence.   

Line 37: Expand the term ECDC.

Line 37-94: Authors need to evaluate the introduction section, may rewrite for better readability. The current introduction is with many numerical figures and may not create requisite interest at first read.

The authors may initially start with little bit interesting detail about hypothesized study without numbers and data.

General Comments: The study provides a comprehensive review of literature especially on structure of antibiotics and mechanism, clinical considerations, and expert opinion. The introduction part needs slight improvement to make manuscript more interesting.

Author Response

Comment: Line 22: Please restructure the sentence.   
Answer: The above-mentioned sentence has been restructured.

Comment: Line 37: Expand the term ECDC.
Answer: The term has been expanded.

Comment: Line 37-94: Authors need to evaluate the introduction section, may rewrite for better readability. The current introduction is with many numerical figures and may not create requisite interest at first read.
Answer: Thank you for your evaluation. The introduction section has been totally revised.

Comment: The authors may initially start with little bit interesting detail about hypothesized study without numbers and data.
Answer: As mentioned, we revised introduction section avoiding too many numbers. Thank you for your suggestions.

General Comments: The study provides a comprehensive review of literature especially on structure of antibiotics and mechanism, clinical considerations, and expert opinion. The introduction part needs slight improvement to make manuscript more interesting.
Answer: Thank you for your positive comments, introduction section has been totally revised in terms of both structure and contents.

Reviewer 2 Report

1. Be sure to define all abbreviations the first time they appear in the document.

2. Respect the reference format within the text and the final list.

3. The in vitro word must be in italics and be homogeneous in their appearance in the document.

4. Be homogeneous in the use of the unit of liter (L) and hour (h).

5. A figure of the chemical structure of molecules can enrich the work.

 6. Add the titles of each figure and table with a brief description to make it more understandable.

7 It would be advisable to add image abbreviations in the figure caption.

8. Figure 3 appears to focus solely on the drug targets of AVI and VAB.

9. A section on side effects on the enteric system could help make clinical decisions to select one or another combination of treatments.

10. It would be necessary to explore a little more about the pharmacodynamic interaction between colistin and fosfomycin with MEM-VAB in section 4.2.

11. The numbering of the methodology section is incorrect.

Author Response

Comment: Be sure to define all abbreviations the first time they appear in the document.
Answer:  We apologize for the above-mentioned typos, we fixed them. We expanded the “ECDC” acronym.

Comment: Respect the reference format within the text and the final list.
Answer:  We checked references format, thank you for your precious advice

Comment: The in vitro word must be in italics and be homogeneous in their appearance in the document.
Answer: Sorry for the typos, we fixed it.  

Comment: Be homogeneous in the use of the unit of liter (L) and hour (h).
Answer: These terms has been revised.

Comment: A figure of the chemical structure of molecules can enrich the work.
Answer: Thank you for the suggestion, we added chemical structures as you suggested.

Comment: Add the titles of each figure and table with a brief description to make it more understandable.
Answer: Thank you for the suggestion, each figure has been revised adding a title.

Comment: It would be advisable to add image abbreviations in the figure caption.
Answer: The figures have been revised.

Comment: Figure 3 appears to focus solely on the drug targets of AVI and VAB.
Answer: The purpose of the figure was to emphasize the differences between the two inhibitors, so the caption has been modified. Thanks for the valuable suggestion.

Comment: A section on side effects on the enteric system could help make clinical decisions to select one or another combination of treatments.
Answer: Thank you for your opinion, we believe it is not worthy adding a whole paragraph about side effects since they were not relevant. In addition, we explain them in the pharmacological section.

Comment: It would be necessary to explore a little more about the pharmacodynamic interaction between colistin and fosfomycin with MEM-VAB in section 4.2.
Answer: There are limited and inconclusive data on the interactions you suggest. Furthermore, the topic of interactions goes beyond the scope of this review. I do not rule out the possibility of writing additional reviews or papers about this subject.

Comment: The numbering of the methodology section is incorrect.
Answer: This mistake has been fixed, thank you for pointing this out.

Reviewer 3 Report

The objective of this narrative review is its clinical aplication. This study explores the pharmacokinetic and pharmacodynamic properties, antimicrobial spectra, in vitro susceptibility testing, and clinical data.

The author should answer this research question in the conclusion. The author should considering CAZ-AVI and MEM- 24 VAB uses carefully. These two antibiotics are antibiotic last resources, only for specific population to hinder antibiotic resistance. (read: AWaRe classification)

Reference:

World Health Organization. (‎2019)‎. The 2019 WHO AWaRe classification of antibiotics for evaluation and monitoring of use. World Health Organization. https://apps.who.int/iris/handle/10665/327957. License: CC BY-NC-SA 3.0 IGO. https://www.who.int/publications-detail-redirect/2021-aware-classification

Even though, this is not a systematic review, the author should write evidenced concisely.

Author Response

Comment: The objective of this narrative review is its clinical application. This study explores the pharmacokinetic and pharmacodynamic properties, antimicrobial spectra, in vitro susceptibility testing, and clinical data.

The author should answer this research question in the conclusion. The author should considering CAZ-AVI and MEM- 24 VAB uses carefully. These two antibiotics are antibiotic last resources, only for specific population to hinder antibiotic resistance. (read: AWaRe classification)

Reference: World Health Organization. (‎2019)‎. The 2019 WHO AWaRe classification of antibiotics for evaluation and monitoring of use. World Health Organization.  

https://apps.who.int/iris/handle/10665/327957. License: CC BY-NC-SA 3.0 IGO. https://www.who.int/publications-detail-redirect/2021-aware-classification. Even though, this is not a systematic review, the author should write evidenced concisely.

Answer: Thank you for your valuable suggestion. We added a final section entitled “key-messages” in which we summarized the conclusions in a schematic way. Hoping it will make the review easier to explain. As regards the attention in administering these new molecules, we have already stated that in the introduction section.